# The size of the EB cap determines instantaneous microtubule stability

Christian Duellberg[1], Nicholas I Cade[1], David Holmes[2†], Thomas Surrey[1]*

[1]Lincoln's Inn Fields Laboratory, The Francis Crick Institute, London, United Kingdom; [2]London Centre of Nanotechnology, London, United Kingdom

**Abstract** The function of microtubules relies on their ability to switch between phases of growth and shrinkage. A nucleotide-dependent stabilising cap at microtubule ends is thought to be lost before this switch can occur; however, the nature and size of this protective cap are unknown. Using a microfluidics-assisted multi-colour TIRF microscopy assay with close-to-nm and sub-second precision, we measured the sizes of the stabilizing cap of individual microtubules. We find that the protective caps are formed by the extended binding regions of EB proteins. Cap lengths vary considerably and longer caps are more stable. Nevertheless, the trigger of instability lies in a short region at the end of the cap, as a quantitative model of cap stability demonstrates. Our study establishes the spatial and kinetic characteristics of the protective cap and provides an insight into the molecular mechanism by which its loss leads to the switch from microtubule growth to shrinkage.

*For correspondence: Thomas.Surrey@crick.ac.uk

Present address: †Sphere Fluidics Limited, Cambridge, United Kingdom

Competing interests: The authors declare that no competing interests exist.

## Introduction

Microtubules are protein filaments found in all eukaryotic cells. They consist of typically 13 protofilaments forming a tube. Their function depends on their ability to switch between growing and shrinking states, called dynamic instability, which is essential for intracellular space exploration, mitosis and migration (*Howard and Hyman, 2009*). The dynamic properties of microtubules are tightly controlled and a target of numerous anti-cancer drugs (*Peterson and Mitchison, 2002*; *Seligmann and Twelves, 2013*). However, the molecular mechanism of microtubule state switching is not understood.

The switch-like nature of microtubule stability is thought to depend on the existence of a stabilising GTP (or GDP-Pi) cap at the end of growing microtubules as a result of addition of GTP-loaded tubulin heterodimers and subsequent GTP hydrolysis and phosphate release (*Carlier, 1982*; *Mitchison and Kirschner, 1984*). However GTP cannot be visualised in individual dynamic microtubules; therefore the cap size and their relation to individual microtubule stability remains unknown. A variety of models exist, assuming either long caps (several tens of tubulin layers, i.e. hundreds of nanometers long, [*Carlier et al., 1984*; *Chen and Hill, 1983*; *Mitchison and Kirschner, 1984*]) or short caps (1-2 tubulin layers, i.e. 8–16 nm only, [*Bayley et al., 1989*; *Bolterauer et al., 1999*; *Caplow and Shanks, 1996*; *Drechsel and Kirschner, 1994*; *Walker et al., 1991*]). Long caps would be a result of random GTP hydrolysis, causing the cap size to increase with growth speed. Short caps would be a consequence of coupled GTP hydrolysis, which would require GTP hydrolysis to accelerate when microtubules grow faster, resulting in a short cap of constant size (*Bayley et al., 1989*). Alternative models exist that assume either currently not observable structural events for loss of stability, such as formation of cracks (*Flyvbjerg et al., 1996*; *Li et al., 2014*; *Margolin et al., 2012*) or defects, or local biochemical criteria at the very end of the microtubule, introducing a distinction between the total GTP cap and its part that is critical for stabilisation (*Bowne-Anderson et al., 2013*; *Gardner et al., 2011b*; *Brun et al., 2009*).

**eLife digest** Much like the skeleton supports the human body, a structure called the cytoskeleton provides support and structure to cells. Part of this cytoskeleton is made up of small tubes called microtubules that – unlike bones – can shrink and grow very quickly. This allows the cell to change shape, move and split into two new cells.

Exactly how the microtubules switch between growing and shrinking was not clear. One suggestion is that a protective cap at the end of microtubule allows it to keep growing and prevents it from shrinking. However, the nature and size of this cap have been debated.

Now, Duellberg et al. have measured the caps of microtubules with high precision by combining the techniques of microfluidics, TIRF microscopy and recently developed image analysis tools. This revealed that the cap sizes change, with longer caps being more stable. In addition, proteins called end-binding proteins can destabilize the cap by binding to it. This allows microtubules to switch from a growing to a shrinking state more often.

Future work could now investigate how changes in cap length cause the microtubules to switch from growing to shrinking. It also remains to be seen whether other proteins also influence the cap to control this switching behaviour.

Proteins of the EB1 family (EBs) transiently bind to growing microtubule ends as their conformation matures in a nucleotide state-dependent manner (*Bieling et al., 2007*; *Kumar and Wittmann, 2012*; *Maurer et al., 2011*), potentially binding to the GTP (or GDP-Pi) cap (*Maurer et al., 2012*; *Seetapun et al., 2012*; *Zhang et al., 2015*). Before microtubules switch from growth to depolymerisation (catastrophe), the size of the EB binding region tends to decrease (*Maurer et al., 2012*), potentially suggesting a link between microtubule stability and the size of the EB binding region. As this region extends over several hundred nanometers, an EB binding site cap would be a 'long cap'.

The recent observation of large microtubule growth fluctuations (*Gardner et al., 2011a*; *Kerssemakers et al., 2006*; *Schek et al., 2007*) challenged 'short cap' models because transient shrinking episodes during growth phases of up to ~ five tubulin layers (which would remove such short caps) did not cause catastrophe (*Schek et al., 2007*). On the other hand, long cap models have been challenged by tubulin dilution experiments (*Voter et al., 1991*; *Walker et al., 1991*). Tubulin dilution causes microtubules to stop growing and hence prevents the addition of new capping tubulins, resulting in microtubules undergoing a catastrophe after a delay of several seconds (*Walker et al., 1991*). This delay time is thought to be the time it takes for the stabilising structure at the microtubule end to disappear and therefore is a measure of momentary stability at the instant of dilution. In these experiments, delay times were reported to be insensitive to the microtubule growth speed, apparently contradicting 'long cap' models that predict higher stability with increasing growth speed due to longer caps (*Padinhateeri et al., 2012*). However, the cap size itself could not be visualised in these previous experiments and no quantitative explanation could be given for the observed values of the delay times. Given the reported contradicting results, the size of the stabilising cap and especially its relationship with microtubule stability is still unclear. Ultimately, this is due to a lack of experiments measuring the protective cap size and relating it to the stability of individual microtubules.

To simultaneously measure momentary microtubule stability and EB cap size with high spatio-temporal resolution, we developed a new microfluidics-assisted two-colour TIRF microscopy assay for fast and complete tubulin removal, combining automated microtubule end tracking and EB protein monitoring. For the first time, this allowed us to measure with close-to-nm and sub-second precision the instantaneous stability of individual microtubules, and the instantaneous size of their EB binding site caps, and to directly investigate their relationship. We find that the EB binding region is indeed the stabilising cap. We demonstrate that microtubules with longer caps are more stable and we present a quantitative kinetic model of momentary microtubule stability. Catastrophe is induced when the number of EB binding sites at the very end of the cap has fallen to 15–30% of its steady state value. This establishes the basic mechanism of catastrophe induction based on measured protective cap properties.

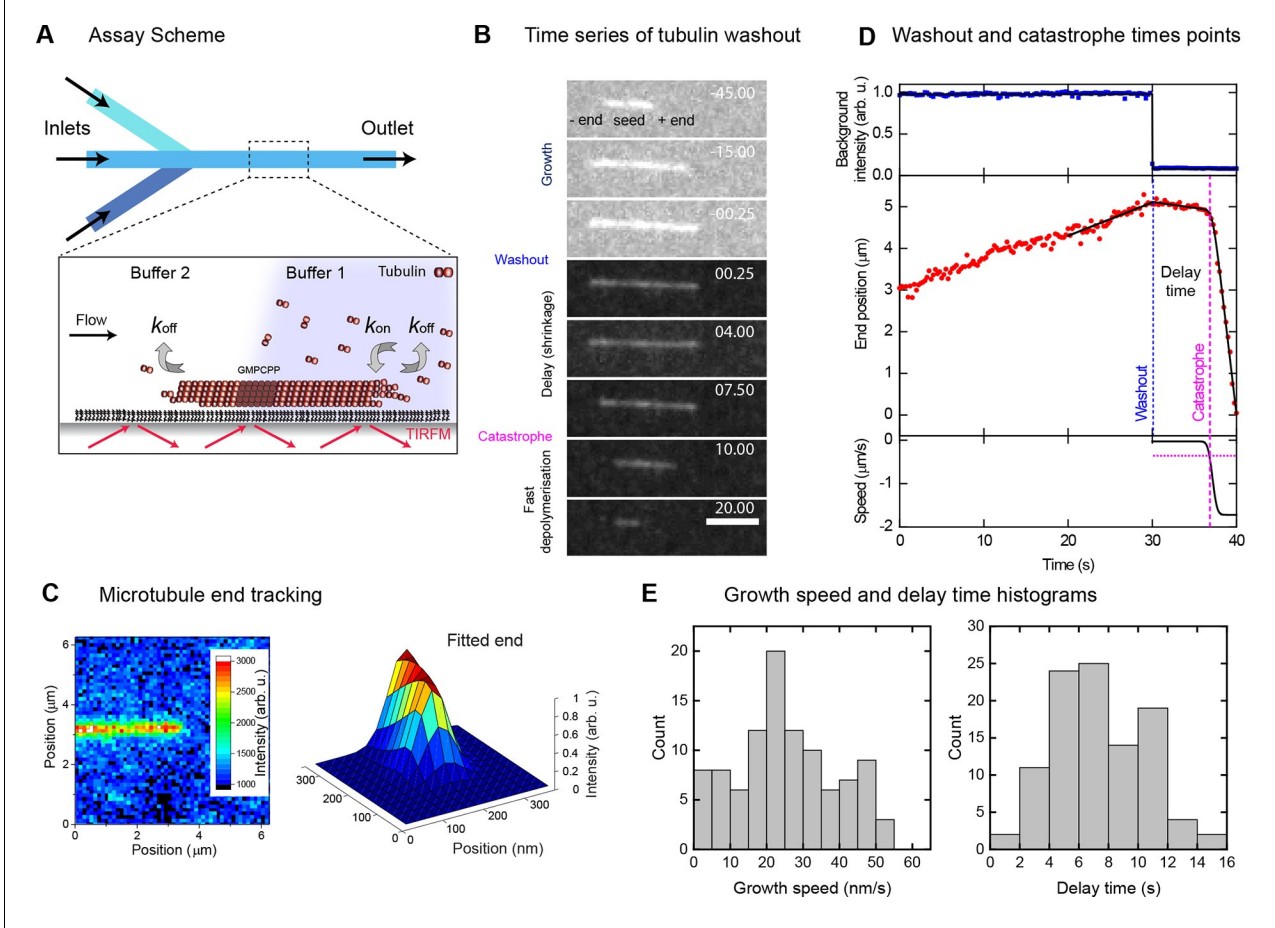

**Figure 1.** Momentary microtubule stability assayed by fast tubulin washout and nm precision plus end tracking. (**A**) Schematic of the microfluidic TIRF microscopy setup. (**B**) TIRF microscopy image sequence of an Alexa568-microtubule before and after washout, with growth, delay and fast depolymerisation periods indicated. Tubulin concentration was changed from 20 to 0 µM at washout. Time in seconds, scale bar is 3 µm. (**C**) Illustration of sub-pixel precision microtubule end tracking using a 2D fitting procedure (Materials and methods, [*Bohner et al., 2015*]). (**D**) Plots of the background fluorescence intensity (top) and end position data (middle) of a washout experiment together with the fits (solid black lines) used to extract the derived parameter values (see Materials and methods). (**E**) Histograms of the growth speeds at tubulin washout (left) and of the subsequent delay times before catastrophe (right) ($n = 101$).

The following figure supplements are available for figure 1:

**Figure supplement 1.** Fast and complete microfluidics-controlled solution exchange.

**Figure supplement 2.** Delay times and microtubule orientations.

## Results

### Microtubule stability increases with growth speed

In a microfluidic device, we immobilised Alexa568-labelled stabilised microtubule seeds on a functionalised glass surface. We then observed their growth in the presence of 20 µM Alexa568-labelled tubulin and GTP by time-lapse total internal reflection fluorescence (TIRF) microscopy (*Figure 1A*, *Figure 1—figure supplement 1A*). Solutions were exchanged within ~200 ms (*Figure 1—figure supplement 1B*); the exchange itself did not affect microtubule growth (*Figure 1—figure supplement 1C & D*). Sudden and complete removal of tubulin stopped growth, and induced a catastrophe after a delay of typically several seconds (*Figure 1B*, *Video 1*), similar to earlier observations after tubulin

**Video 1.** Tubulin washout experiment, corresponding to *Figure 1*. A drop in background fluorescence marks the time point, when tubulin is removed. Microtubules undergo catastrophe with a delay. Time is in seconds, where 0 is set to the washout time point.

dilution (*Walker et al., 1991*). This delay is thought to be caused by the temporary survival of the stabilising cap.

We imaged 101 microtubules in the presence of 20 µM tubulin at 4 Hz and tracked their plus ends with a precision of ~30 nm (*Figure 1C*). Fitting traces of the end position and fluorescence background (Materials and methods) allowed us to determine the washout and catastrophe times with sub-sampling time precision, as well as the instantaneous growth speeds measured over a 10 s time period just before washout (*Figure 1D*, *Figure 1—figure supplement 1B*). Growth speeds varied considerably around a mean of 28 nm/s (*Figure 1E*, left). The delay times between washout and catastrophe also showed a broad and non-exponential distribution, with a mean of 7.3 s (*Figure 1E*, right), similar to previous dilution experiments (*Walker et al., 1991*). As bending can affect the material properties of microtubules (*Schaedel et al., 2015*), we tested whether the measured delay times were influenced by mechanical stress, potentially induced by microtubule bending in our assay. We determined the orientation of the growing microtubule end regions relative to the flow direction before tubulin washout, and found that the variation of orientations was relatively small (mean orientation 3.1° with a standard deviation of 7.3°) indicative of good microtubule alignment. No correlation between the delay time and the magnitude of the orientation was observed (*Figure 1—figure supplement 2A*, blue data), indicating that mechanical stress is not responsible for the observed variations of the momentary microtubule stabilities in our assay.

However, in contrast to a previous report (*Walker et al., 1991*), the delay times clearly increased with increasing growth speed, provided that speeds were measured directly before washout (*Figure 2A*, *Figure 2B* left, Spearman's rank correlation coefficient $\rho = 0.69$, $p<10^{-15}$). This demonstrates that microtubules are more stable when they grow faster. Interestingly, the strong correlation decreased when the growth speed was measured at earlier times before tubulin washout (*Figure 2A, B* right), as quantitatively demonstrated by a gradual decrease of the correlation coefficient and a gradual increase of its p-value (*Figure 2C*). Therefore, the measured delay times reflect microtubule stability at the moment of tubulin washout and this stability varies over time as the growth speed fluctuates. This might explain why previous tubulin dilution experiments failed to detect the correlation between delay times and growth speeds; there average growth speeds were measured over long time intervals (*Walker et al., 1991*).

## Slow shrinkage for hundreds of nanometers before catastrophe excludes short cap models

A mean microtubule end trajectory was generated by averaging all tracks after washout, aligned with respect to the catastrophe time (*Figure 2D*). This average track demonstrated that microtubules did not simply pause between washout and catastrophe as previous lower resolution measurements indicated (*Walker et al., 1991*); instead microtubules shrank slowly on average at 26 nm/s before catastrophe. This compared to a much faster depolymerisation speed of 480 nm/s after catastrophe. Our resolution allowed us to also measure slow shrinkage at the level of individual microtubules (*Figure 2E* top), yielding a similar mean shrinkage speed (*Figure 2—figure supplement 1C*). The observed slow shrinkage agrees with the established view that in the presence of tubulin, growth results from the difference of fast tubulin association and slower dissociation events (*Gardner et al., 2011a*; *Walker et al., 1988*); we observe the latter here directly for the first time. We also measured the nanometer-range distances that microtubules shrank between washout and catastrophe. These shrinkage lengths also tended to increase with growth speed (*Figure 2E* bottom, $\rho = 0.51$, $p<10^{-7}$), supporting the idea that faster microtubules are more stable, apparently due to longer stabilising caps. The mean shrinkage length was 220 nm, i.e. more than 25 tubulins per protofilament. This directly excludes models based on short stabilising caps.

Interestingly the observed shrinkage length before catastrophe was in the same range as previously reported 'comet lengths' of the EB binding regions in cells (*Seetapun et al., 2012*) and in vitro (*Bechstedt et al., 2014*; *Bieling et al., 2007*; *Dixit et al., 2009*), further supporting the possibility of

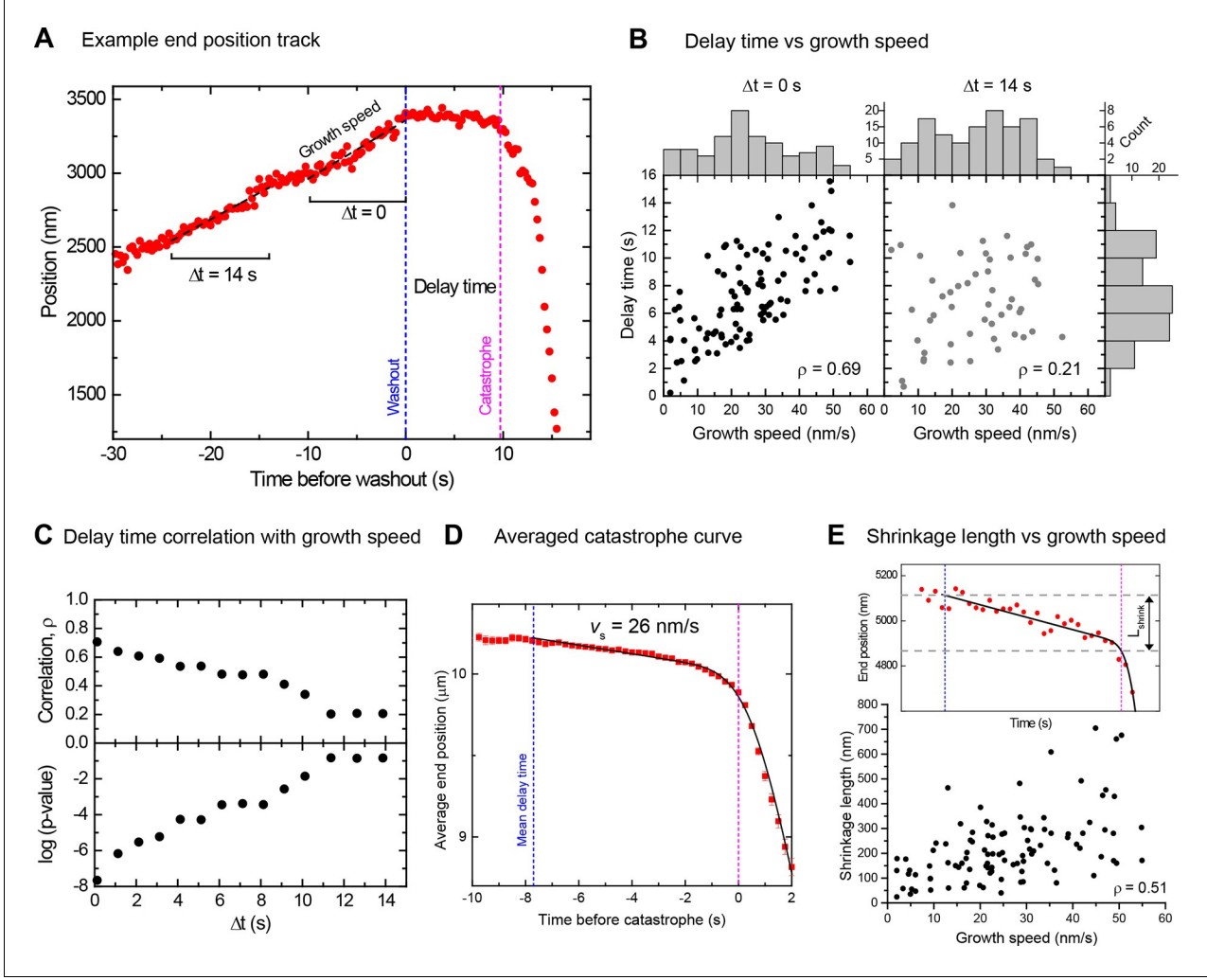

**Figure 2.** Momentary stability increases with growth speed at washout time. (**A**) Example end position-time plot. Growth speeds were measured by linear fits over a 10 s time window directly before washout ($\Delta t = 0$) or for comparison also at earlier times up to 14 s before washout. (**B**) Scatter plots showing a positive correlation between delay times and growth speeds when measured directly before washout ($\Delta t = 0$); correlation strength (Spearman's rank correlation coefficient $\rho$) is significantly reduced for $\Delta t = 14$ s. Histograms of growth speeds (top) and delay times (right) are also shown. (**C**) Spearman's rank correlation coefficient $\rho$ (top) and corresponding p-values (below) between delay time and growth speed indicate that the correlation strength decreases progressively when the time window over which speed is determined is shifted away from the washout time point. (**D**) Averaged microtubule end position trace after washout (alignment with respect to catastrophe times). Errors are s.e.m. (**E**) Top: Example trace of a shrinkage episode for an individual microtubule after washout, with fit (Materials and methods) illustrating the definition of shrinkage length. Blue and magenta lines indicate washout and catastrophe times, respectively. Bottom: scatter plot of the shrinkage lengths versus growth speeds.

The following figure supplement is available for figure 2:

**Figure supplement 1.** Determination of catastrophe and shrinkage parameters after tubulin washout.

a link between the EB binding region and microtubule stability, as suggested recently (*Maurer et al., 2012*).

## Binding of Mal3-GFP to the EB binding region decreases momentary microtubule stability

To visualise the EB binding region before and after tubulin washout, we added 200 nM GFP-labelled Mal3, the fission yeast EB (*Beinhauer et al., 1997*; *Busch and Brunner, 2004*), keeping its concentration constant throughout the tubulin washout experiment (*Figure 3A*, *Video 2*). EBs do not only report on the size of their binding region, but also affect the kinetic processes determining its size:

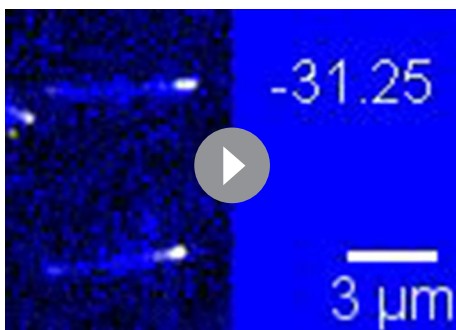

**Video 2.** Tubulin washout experiment in the presence of 200 nM Mal3-GFP (yellow), corresponding to *Figure 3*. Microtubules are depicted in blue. The microtubule with the higher Mal3-GFP signal at the time point of washout exhibits a longer delay until catastrophe occurs. Time is in seconds, where 0 is set to the washout time point. To enhance clarity of presentation, background subtraction (imageJ, 30 pixel rolling ball plug-in) was applied to suppress the large difference between total intensities in the microtubule channel before and after washout.

they mildly increase microtubule growth speeds and accelerate the turnover of their own binding sites, effectively shortening the region they bind to (*Maurer et al., 2011*; *Maurer et al., 2014*). Analysis of 139 microtubules showed that after tubulin washout, the mean delay time was now reduced approximately twofold to 3.5 s (*Figure 3B*), despite a mild increase of the growth speed to 33 nm/s (*Figure 3B*, inset). Hence, binding of Mal3-GFP reduces the momentary microtubule stability, reminiscent of the EB-induced stimulation of catastrophes at steady state (*Komarova et al., 2009*; *Maurer et al., 2014*; *Mohan et al., 2013*). This suggests that the acceleration of microtubule end maturation that shortens the EB binding region is, at least in part, responsible for decreased momentary microtubule stability.

## Microtubules with larger EB site caps are more stable

At the moment of tubulin washout the Mal3-GFP intensities at microtubule end regions varied considerably (*Figure 4A*, blue histogram). At a constant Mal3-GFP concentration the fraction of occupied binding sites is constant; additionally, the Mal3-GFP binding/unbinding turnover is fast compared to other processes (*Bieling et al., 2007*): hence, the observed variation of the Mal3-GFP intensities results mainly from variations of the EB binding site numbers at different microtubule ends (*Figure 4—figure supplement 1*). Higher Mal3-GFP intensities were observed for faster growth speeds ($\rho = 0.74$, $p<10^{-15}$) (*Figure 4B*). This indicates that individual microtubules that happened to be growing faster at the moment of tubulin washout had longer EB site caps, under otherwise identical conditions. This extends previous observations made at the ensemble level showing that the mean EB comet length increases with mean growth speed when varying the tubulin concentration (*Bieling et al., 2007*).

Immediately after tubulin washout the Mal3-GFP intensity in the microtubule end region started decreasing (*Figure 4C*, green traces). Strikingly, microtubules with initially smaller EB site caps displayed shorter delay times than those with larger caps (*Figure 4C*, compare top vs bottom), as demonstrated quantitatively by a strong correlation between delay times and initial Mal3-GFP end intensities ($\rho = 0.78$, $p<10^{-30}$) (*Figure 4D* top, *Figure 4—figure supplement 2*). Consistently, the more stable microtubules were again the faster growing ones ($\rho = 0.66$, $p<10^{-15}$) (*Figure 4E*, *Figure 4—figure supplement 2C* top). Both correlations were strongest when the EB cap size and the growth speed were measured directly at tubulin washout; the correlation coefficients decreased and p-values increased for growth velocities and Mal3-GFP intensities measured at earlier times before washout (*Figure 4F*, *Figure 4—figure supplement 2*). This demonstrates again that microtubule stability fluctuates over seconds as the growth state and the size of the EB binding region fluctuate. At the time point of catastrophe, the total Mal3-GFP intensity, and hence the size of the EB site caps, was on average reduced to 29% of the initial size (*Figure 4D*, compare top vs middle, considering background shown in bottom). The similarity of the observed reduction in size of the EB binding regions at the moment of catastrophe suggests that catastrophe occurs when a critical threshold of EB sites is reached.

## Microtubule end maturation and slow shrinkage gradually remove the protective cap after stop of growth

We averaged both microtubule end trajectories and Mal3-GFP intensity traces after tubulin removal, all aligned with respect to the catastrophe time. The mean Mal3-GFP intensity decay was roughly mono-exponential until catastrophe, with a measured rate of 0.33 s$^{-1}$ (*Figure 5A*). This rate could be

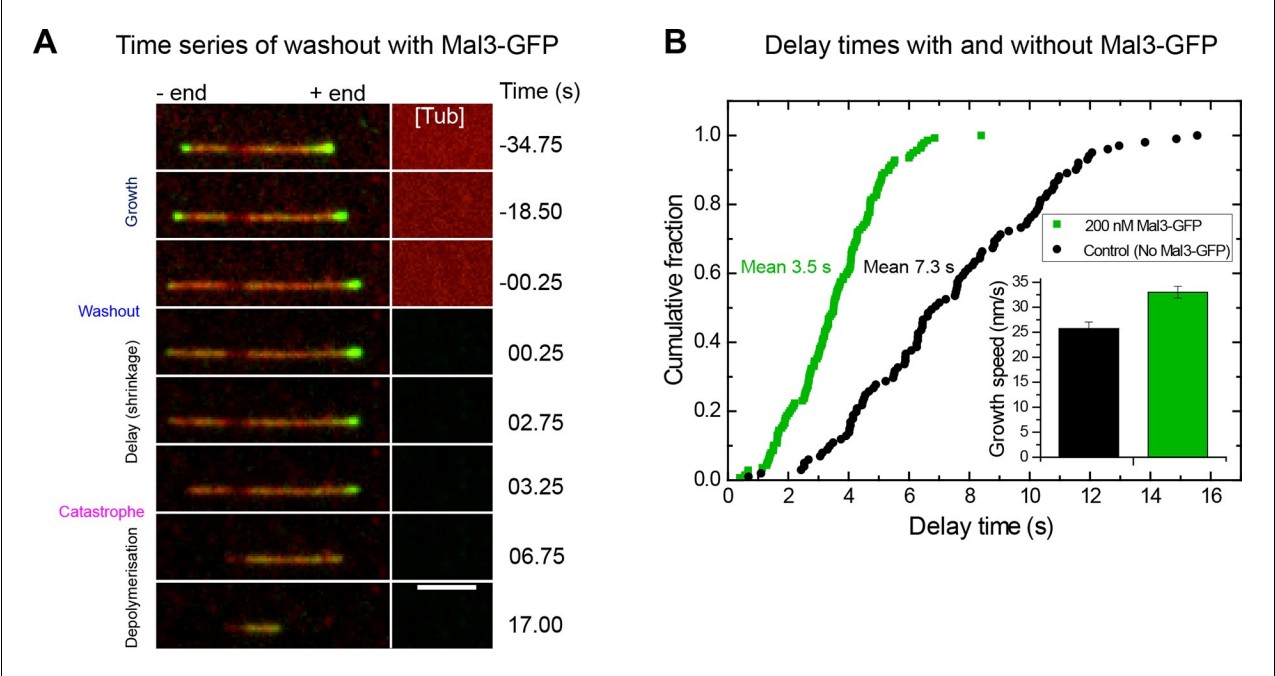

**Figure 3.** Mal3 shortens the delay between tubulin washout and catastrophe. (**A**) TIRF microscopy image sequence of a Alexa568-microtubule (red) in a tubulin washout experiment in the constant presence of 200 nM Mal3-GFP (green). Tubulin was present at 20 μM before washout. Background has been subtracted; the raw background intensity of the tubulin/microtubule channel is depicted on the right hand side ([Tub]), indicating when tubulin is removed; other conditions are as in *Figures 1* and *2*. Time in seconds, scale bar is 3 μm. (**B**) Cumulative delay time distributions in the presence of 200 nM Mal3-GFP (green) and in its absence (black). Inset: bar graph of the corresponding mean growth velocities before washout; error bars are s.e.m.

quantitatively explained as a consequence of two kinetic processes removing the cap (*Figure 5B*): (i) slow shrinkage of the microtubule after washout (mean $v_s$ = 40 nm/s in the presence of Mal3-GFP) which removes tubulins from the end; (ii) continued microtubule maturation after washout which transforms EB binding sites everywhere in the cap into mature lattice ($k_m$ = 0.16 s$^{-1}$). The latter can be determined from the analysis of averaged Mal3-GFP intensity profiles (comet analysis) (*Figure 5C*, *Supplementary file 1A*, Materials and methods) (*Maurer et al., 2014*). Together with the measured mean growth speed $v_g$ = 33 nm/s before washout, these known kinetic parameter values predict a theoretical decay rate for the size of the EB cap of $k_{Mal3}$ = $k_m(v_s/v_g+1)$ = 0.35 s$^{-1}$ (Materials and methods), in good agreement with the measured value of 0.33 s$^{-1}$.

Knowing the rate of the Mal3-GFP end intensity decay and the relative size of the EB binding region at catastrophe compared to the moment of tubulin washout, the theoretically expected delay time can be calculated (*Figure 5D* left, Materials and methods), assuming that catastrophe is reached when 29% of the EB cap is left under the conditions studied here (same data as in *Figure 4*). This demonstrates that the mean delay time is determined by the growth characteristics of the microtubule (growth speed $v_g$ and shrinkage speed $v_s$) and by the kinetics of microtubule end maturation (maturation rate $k_m$). Similarly the mean shrinkage length (165 nm in the presence of Mal3-GFP, or ~20 tubulin lengths) could also be calculated (*Figure 5D* right, *Figure 2—figure supplement 1B*, Materials and methods). Furthermore a rough estimate of the standard deviations of the delay times and the shrinkage lengths can be obtained assuming that they are mostly determined by the observed variations of the size of the EB binding region at tubulin washout and catastrophe (*Figure 5D*, Materials and methods). The good agreement between the measured and predicted delay times and shrinkage lengths shows that the temporal and spatial scale of the response of the microtubules to a sudden stop of growth can be quantitatively explained.

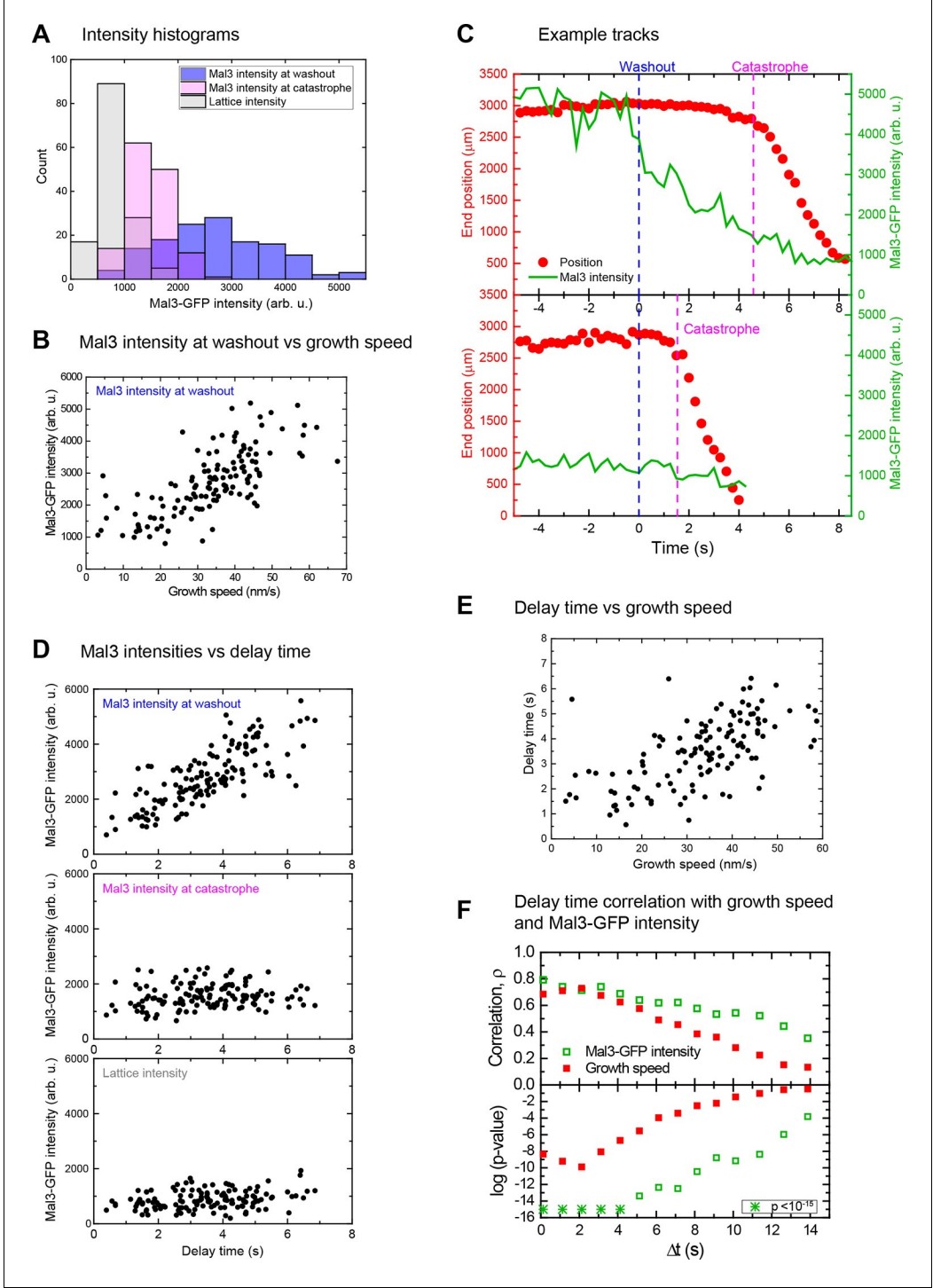

**Figure 4.** Microtubules with longer EB site caps are more stable. (**A**) Histograms of the Mal3-GFP intensity at microtubule ends at the moment of tubulin washout (blue) and catastrophe (magenta) (*n* = 139). The corresponding lattice intensity at tubulin washout (grey) is also shown. Mal3-GFP was present at 200 nM throughout the experiment and tubulin at 20 µM before washout. (**B**). Scatter plot showing the positive correlation of Mal3-GFP intensities and growth speeds determined directly before washout. (**C**) Two representative end position time plots (red dots) with the corresponding Mal3-GFP signals (green line). The microtubule with an initially stronger Mal3-GFP signal at washout (top) has a longer delay time until catastrophe compared to the microtubule with weaker Mal3-GFP signal at washout (bottom). (**D**) Scatter plots of Mal3-GFP intensities at microtubule ends at the time of tubulin washout (top), at microtubule ends at the time of catastrophe (middle), and on the microtubule lattice (1.5 µm from the microtubule end) (bottom) versus the delay times. At the moment of catastrophe, on average 29% of the EB binding sites present at tubulin washout were left, calculated by comparing the Mal3-GFP intensity at washout and catastrophe, from each of which the lattice intensity was subtracted. (**E**) Scatter plot

*Figure 4 continued on next page*

*Figure 4 continued*

showing the positive correlation of delay times and growth speeds determined directly before washout. Delay times were independent of small variations of the microtubule orientation (*Figure 1—figure supplement 2A*, green data). (F) Decrease of correlation strength ρ between Mal3-GFP intensities and delay times (open green symbols) or growth speeds and delay times (solid red symbols) when the intensities/speeds were measured at earlier time points before washout, similar to *Figure 2*. Spearman's rank correlation coefficients ρ (top) and corresponding p-values (bottom, log scaled) are depicted.

The following figure supplements are available for figure 4:

**Figure supplement 1.** The main source of Mal3-GFP intensity fluctuations at microtubule ends growing in the presence of GTP are fluctuations of the size of the EB binding site cap.

**Figure supplement 2.** The instant EB cap length defines momentary stability.

## A minimal cap density close to the microtubule end - but not the total number of cap sites - is required for microtubule stability

We observed that a large part of the EB binding region is lost at catastrophe, however part of it is still present when catastrophe occurs. This suggests that a threshold may have to be reached to induce catastrophe, raising the question of what exactly constitutes this threshold. Two extreme possibilities can be envisaged: A minimal total number of cap sites anywhere in the entire cap might be required for stability. Alternatively a minimal density of cap sites only at its very end where the cap site density is highest might be needed for stability of the cap. In other words, either the entire cap or only its highest density region could be critical for stability. These two scenarios predict different dependencies of the delay times on the growth speed and hence cap length (Materials and methods). To explore the momentary microtubule stabilities over a larger range of cap sizes, we performed tubulin washout experiments at a range of different tubulin concentrations from 10 µM to 35 µM, and at an increased magnesium ion concentration to further increase the velocity range (*O'Brien et al., 1990*). In total 210 microtubules were analysed. As the growth speed distributions at different tubulin concentrations overlapped strongly (*Figure 6A*, top), we speed-sorted the data when calculating averages as a function of speed (*Figure 6—figure supplement 1A*) (*Maurer et al., 2014*). Average delay times extracted for seven speed groups from over 200 individual microtubule tracks displayed the expected positive correlation between delay times and growth speeds, however showing a weaker dependence especially for the higher speed range (*Figure 6A* bottom). This correlation was masked when displaying the delay times simply as a function of tubulin concentration (*Figure 6—figure supplement 1B*), again emphasizing the importance of correlating delay times with momentary speeds in the presence of growth fluctuations.

Together with the measured end maturation rate (*Supplementary file 1A*), growth and shrinkage speeds (*Figure 6—figure supplement 1C*) for this condition, the dependence of the mean delay times on growth speed could be quantitatively explained by assuming that catastrophe is induced when a critical 'end density' of cap sites (*Figure 6B*, $n_{crit}$) is reached in the decaying cap. A fit (*Figure 6A*, bottom, solid red line) (Materials and methods) predicts that independent of the initial length of the stabilising cap, catastrophe occurred when around a fifth of the initial density of cap sites ($f_{crit} = 0.20$) was left at the end of the decaying cap, i.e. ~3 per tubulin layer. The longer the caps at washout, i.e. the faster microtubules grow, the less dependent the momentary microtubule stability becomes on cap length. This is because EB site maturation becomes more important relatively to end shrinkage for reaching the 'end density' threshold, with increasing cap length at washout. In contrast, an alternative model assuming that catastrophe occurs when a critical 'total number' of sites anywhere in the cap (*Figure 6B*, $N_{crit}$) is reached did not agree with the data (*Figure 6A*, dashed line): this model predicts a stronger increase of delay times with growth speed for longer caps as a consequence of increasing total EB binding site numbers with cap length at washout.

Fits to the other speed-sorted mean delay times as a function of mean growth speed (from data shown in *Figure 2B* left and *Figure 4E*) produced also good agreement with the data for a simple critical 'end density' threshold (*Figure 6C*). This analysis revealed furthermore that the addition of Mal3-GFP destabilised the cap by increasing the threshold from 14% to 28% of the initial end density, i.e. from ~2 to ~4 cap sites per tubulin layer (*Supplementary file 1B*).

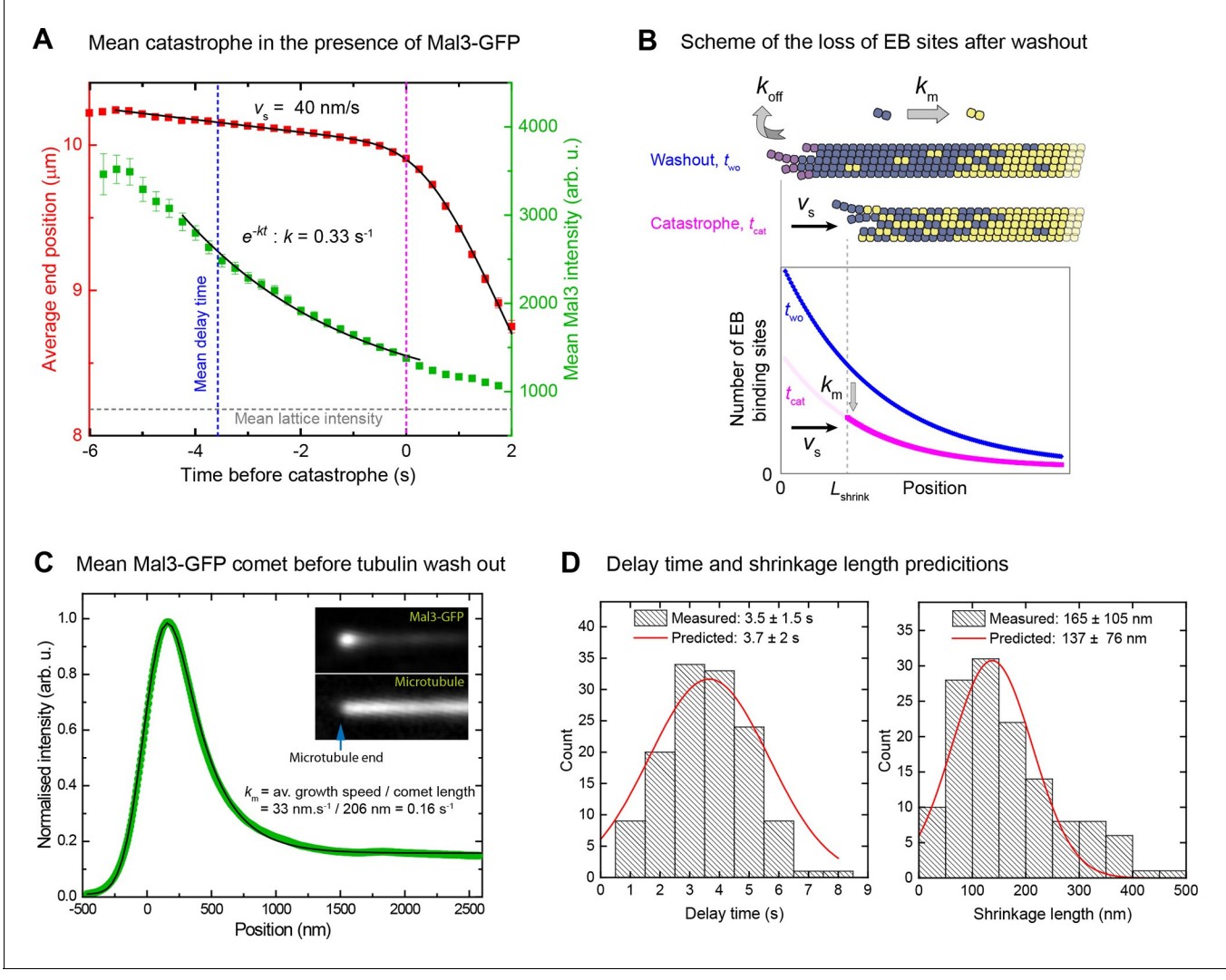

**Figure 5.** When an EB cap size threshold is reached catastrophe is induced. (**A**) Averaged microtubule end position traces (red) and Mal3-GFP intensity traces (green) after tubulin washout (alignment with respect to the catastrophe time), and fits (black lines) (Materials and methods). Same data set as in *Figure 4*. (**B**) Top: Scheme illustrating the two processes leading to EB cap loss after tubulin washout: disassociation of tubulin from the end ($k_{off}$) resulting in slow shrinkage ($v_s$), and maturation of EB binding sites into mature lattice ($k_m$). Together these processes define the kinetics of the decay of EB binding sites (Materials and methods). Bottom: simple model of average distributions of the EB binding sites at washout ($t_{wo}$) and catastrophe ($t_{cat}$) times. (**C**) Comet analysis: Average (green) of 5500 Mal3-GFP intensity profiles aligned with respect to the microtubule end position as described in (*Maurer et al., 2014*) and a fit to the data (black) yielding the comet length, which together with the growth velocity provides the maturation rate (Materials and methods). (**D**) Histograms of the distribution of measured delay times (left) and shrinkage lengths (right) overlaid with theoretical estimates (red solid line) from an empirical kinetic model (Materials and methods) based on the measured GFP intensity distributions (*Figure 4A*) and the measured kinetic parameters from panel A and C.

To gain insight into the length along the microtubule end over which a critical minimal density of cap sites is required for stability, we examined the data using a third version of the simple maturation model: this explicitly considered the number of cap sites within an extended but finite distance of the shrinking end (*Figure 6D*, inset). Good global fits to our complete dataset, consisting of all delay times and also the Mal3-GFP intensities at washout and catastrophe, were obtained for fixed end region lengths ranging between 8 and ~80 nm (*Figure 6D*, *Figure 6—figure supplement 2*). This puts an upper limit on the size of the critical part of the cap over which a minimal density needs to be maintained for microtubule stability.

For simplicity we have assumed here throughout that the EB binding site region starts directly at the microtubule end (one-step end maturation). Previously, a detailed analysis of fluorescent EB end

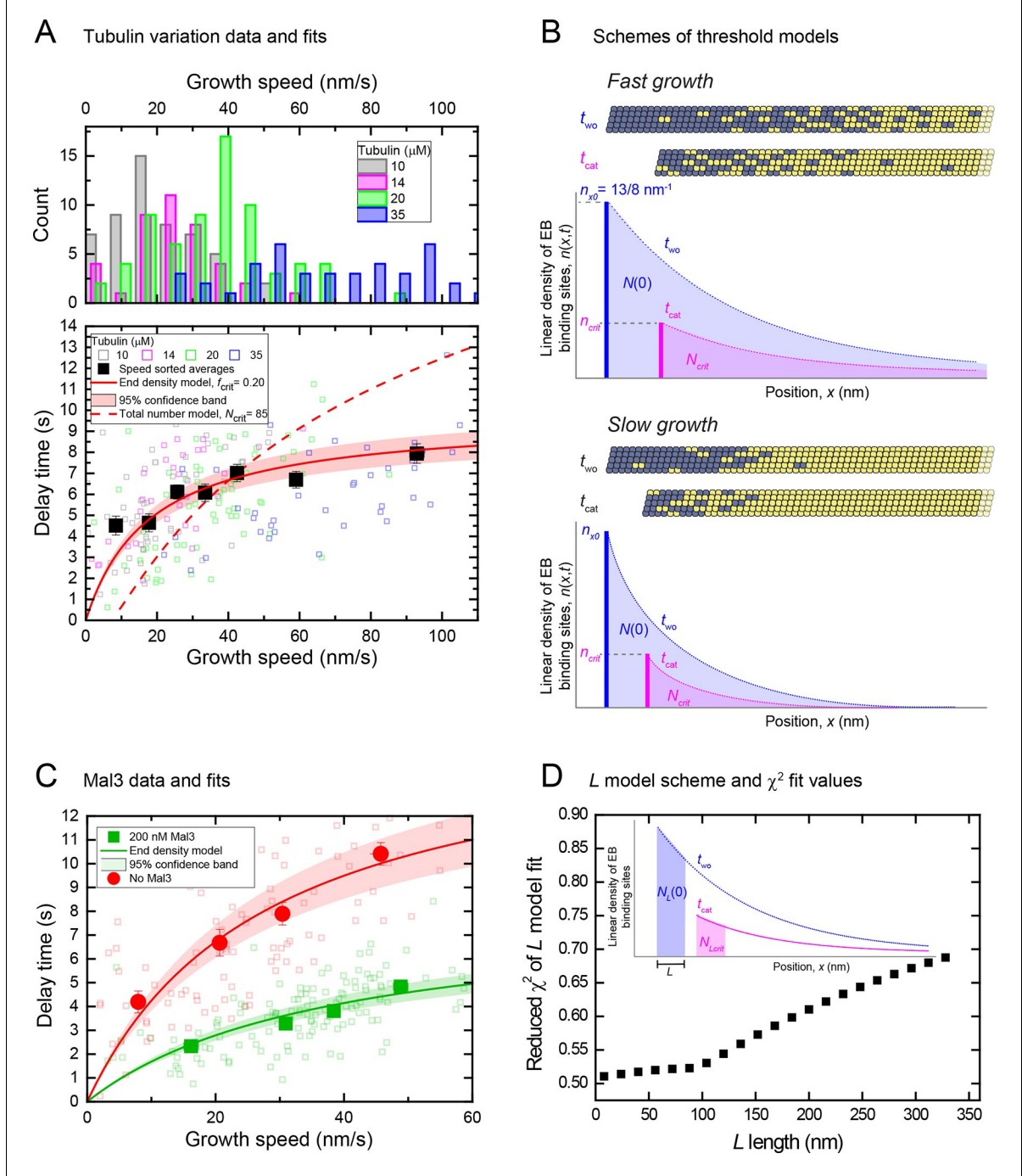

**Figure 6.** An end density threshold explains the dependence of the mean delay times on growth speed. (A) Top: Histograms of overlapping growth speed distributions at tubulin washout for 10, 14, 20 and 35 μM tubulin. Bottom: Scatter plots of the corresponding individual delay times versus growth speeds (open symbols) and of the averages for 7 different speed groups (filled symbols) (see *Figure 6—figure supplement 1A*). Error bars are s.e.m. *n* = 210. Fits are shown for kinetic models assuming that catastrophe is induced when either a critical EB binding site density in the end region (solid line, *Equation 11* in Materials and methods) or a critical total number of EB sites (dashed line, *Equation 13* in Supplemental Materials and methods) is reached (Materials and methods). As for the other datasets, delay times were independent of small variations of the microtubule orientation (*Figure 1—figure supplement 2B*). (B) Schemes illustrating the two threshold scenarios for a fast (top) and slowly (bottom) growing microtubule. The total number of EB binding sites (shaded areas under the curves) is different for the two speeds when the same critical end density of sites (bold vertical lines indicating the maximum amplitude of the curves) has been reached. This illustrates that the threshold values of a kinetic model assuming a critical 'end density' threshold and of a model assuming a critical 'total number' threshold have different dependencies on the initial length of the EB binding region before tubulin washout. This leads to different predictions of the dependence of the delay times on the growth speed as shown in (A).

*Figure 6 continued on next page*

*Figure 6 continued*

(C) Scatter plots of the data previously shown in *Figure 2B* left and *Figure 4—figure supplement 2C* top, without and with Mal3-GFP (green and red symbols, respectively): individual delay times versus growth speeds (open symbols) and of the averages for 4 different speed groups (filled symbols). Error bars are s.e.m. Solid lines are fits to the data using the end density threshold model (*Equation 11* in Materials and methods). The delay times of the data without Mal3 differ slightly from those shown in (A) as a consequence of different Mg concentrations. (D) Global fits to the speed-sorted data in A, C and the speed-sorted Mal3-GFP intensities at tubulin washout and catastrophe (from *Figure 4D* top and middle) using the L- model that considers as a threshold the number of EB cap sites within a region of length L behind the microtubule end. Reduced $\chi^2$ values are shown for a range of L values. Example fits at $L = 8$ nm, 80 nm and 320 nm are shown in *Figure 6—figure supplement 2*. Inset: Scheme showing the region considered in the L model and the number of EB sites within that region (Materials and methods).

The following figure supplements are available for figure 6:

**Figure supplement 1.** Growth and shrinkage speeds and delay times of the data sets with varied tubulin concentrations.

**Figure supplement 2.** Threshold model fits considering the number of binding sites within a specific length, *L*.

**Figure supplement 3.** Threshold model fits considering 2-step maturation of the microtubule end.

profiles has revealed an additional small non-binding region at the very end of the microtubule before the actual EB binding region (*Maurer et al., 2014*). This region could be accounted for by a two-step end maturation process consisting first of fast generation of EB binding sites, and a subsequent slower maturation into lattice sites. Applying this more complex model here did not improve the quality of the fits (*Figure 6—figure supplement 3*) and confirmed that the threshold of stability is defined by a critical end density of cap sites and not by a critical total number of cap sites (*Figure 6—figure supplement 3A*). In the absence of Mal3, the 2-step maturation model predicted threshold values for the end density that were ~25% larger compared to the simpler 1-step model. In the presence of Mal3 the two models predicted the same threshold value (within error) due to the first maturation step being very fast in the presence of EB1 family proteins (*Maurer et al., 2014*). Therefore, the conceptually simpler end maturation model is sufficient to capture the basic principles determining momentary microtubule stability, especially for the more physiological condition in the presence of an EB1 family protein.

## Discussion

### Momentary microtubule stability can be monitored in real time

We have performed tubulin washout experiments with high spatial and temporal resolution. By simultaneously monitoring the size of the EB binding region we provide compelling evidence that the protective cap is the EB binding region. Previously, it was noted that the size of the EB binding region decreased before catastrophe at steady state (*Maurer et al., 2012*). Here, using sudden tubulin removal, we directly investigated for the first time the relationship between the size of the protective cap and the momentary stability of individual microtubules. We found that faster growing microtubules have longer EB caps and are more stable after tubulin washout. Therefore, our results provide the first direct experimental demonstration for the original proposal that microtubule stability would increase with growth speed as a consequence of longer protective caps (*Carlier et al., 1984*; *Mitchison and Kirschner, 1984*).

### The protective cap is long

Our observations resolve the previous contradiction between tubulin dilution experiments, where delay times had been reported to be independent of growth speeds (*Voter et al., 1991*; *Walker et al., 1991*), and steady state lifetime measurements, where longer life times were observed for faster growth (*Gardner et al., 2011b*; *Janson et al., 2003*; *Walker et al., 1988*). This apparent discrepancy led to the development of various 'short cap' models of microtubule stability or to more elaborate models postulating currently non-observable structural defects or cracks as being critical for catastrophe induction (*Bolterauer et al., 1999*; *Bowne-Anderson et al., 2013*; *Brun et al., 2009*; *Flyvbjerg et al., 1996*; *Li et al., 2014*; *Margolin et al., 2012*; *Piette et al.,*

*2009*). However, these models are inconsistent with the dependence of microtubule stability on instantaneous growth speed (*Figure 2B*, *4E* and *6A*) and are incompatible with the measured slow shrinking phase after tubulin washout removing ~25 tubulin layers on average from microtubule ends before catastrophe onset (*Figure 2* and *5*). These observations and the increase of steady state microtubule lifetimes (*Gardner et al., 2011b*; *Janson et al., 2003*; *Walker et al., 1988*) with increasing growth speed as well as the recently observed shrinking episodes during steady state growth (*Schek et al., 2007*) are all consistent with the view that the protective cap is hundreds of nanometers long and that a random process limits its length as proposed early on for the 'random GTP hydrolysis' model (*Carlier et al., 1984*; *Mitchison and Kirschner, 1984*; *Padinhateeri et al., 2012*).

Whether this random process is GTP hydrolysis or rather phosphate release remains still to be determined. Given the high affinity of EBs to berylliumfluoride-microtubules (often used as a GDP-Pi mimic [*Carlier et al., 1989*]) and GTPγS microtubules, but not to GMPCPP microtubules (thought to be the canonical GTP analogue for microtubules [*Alushin et al., 2014*; *Hyman et al., 1992*]) it is plausible that both the GTP and GDP-Pi nucleotide states are stabilising, with GDP-Pi tubulins forming the majority of the cap to which EBs bind with high affinity, consistent with the EB binding pattern along microtubule ends (*Maurer et al., 2011*; *2014*; *Zhang et al., 2015*).

## Microtubule stability fluctuates

We discovered that the correlations between microtubule growth speed, EB cap length and momentary microtubule stability (delay time after tubulin removal) were lost when growth speed or EB cap size were measured several seconds before tubulin washout (*Figure 2,4*, *Figure 4—figure supplement 2*). This indicates that the momentary microtubule stability fluctuates as growth speed varies and that EB caps store a 'memory of stability' over the time scale of several seconds, probably as a consequence of the lifetimes of the EB binding sites being in this range. The observation of this memory excludes memoryless models of microtubule dynamics (*Bayley et al., 1989*; *Bolterauer et al., 1999*; ). It also explains why lower correlations between mean growth speeds measured over longer time intervals and momentary microtubule stability were observed in previously experiments with lower resolution (*Walker et al., 1991*), emphasizing the dynamic nature of microtubule stability, being a prerequisite for dynamic instability.

## Loss of cap stability

Our study shows that the entire EB binding site region serves as the stabilising cap. In space, its stability decreases gradually with binding site density. Large parts of the cap can be lost before it loses stability, strictly excluding cap models where the entire cap size is small (*Bayley et al., 1989*; *Bolterauer et al., 1999*; *Bowne-Anderson et al., 2013*). The total cap is however much larger than what is required for its stability. This property allows the cap to withstand strong growth fluctuations that include even short depolymerisation episodes (*Schek et al., 2007*).

The observed delay times after tubulin washout were more sensitive to growth speed variations in the lower growth speed regime (*Figure 6A*). Kinetic threshold models revealed that this behaviour can be accounted for if only the highest density region of the cap, i.e. its part closest to the microtubule end, determines its stability. This agrees with earlier conclusions that the minimal critical cap required for stability is short (*Caplow and Shanks, 1996*; *Drechsel and Kirschner, 1994*). We found that for the microtubule to be stable on average 15–30% cap sites needed to be left in the end region of the cap which had a length of up to ~10 tubulin layers at most. This conclusion drawn from our tubulin washout experiments is in good agreement with previous observations during steady state growth for the reduction of EB binding sites before catastrophes (*Maurer et al., 2014*), demonstrating that it is independent of the specifics of the tubulin washout experiment.

We observed considerable variation of individual delay times around their average values indicating the influence of additional stochastic effects that our simple kinetic threshold model does not capture. Such variability can be expected to result for example from differences in the exact spatial distributions of stabilising tubulins in the cap region and from the structural consequences these different distributions have for stability. Nevertheless our model captures the fundamental characteristics of momentary microtubule stability and provides a quantitative link with the kinetic properties of microtubule growth and GTPase-linked microtubule end maturation.

At steady state, the catastrophe frequency has been reported to depend on the time of microtubule growth, i.e. on the microtubule age (*Gardner et al., 2011b*; *Odde et al., 1995*). Here we analyzed only microtubules which grew for similar time periods before tubulin washout. This allowed us to neglect ageing effects in our analysis. Currently there is no agreement on the mechanistic origin of microtubule ageing under steady state conditions (*Bowne-Anderson et al., 2013*; *Coombes et al., 2013*; *Zakharov et al., 2015*). In the future, tubulin washout experiments as presented here can be expected to provide valuable novel insights into the effects of ageing on microtubule stability which will then likely require an extension of our current model of the momentary microtubule stability.

## Destabilising the protective cap

Here, we gained also more mechanistic insight into how binding of EB proteins to the protective cap region influences microtubule dynamics. EBs destabilize microtubules in three ways: (i) acceleration of conformational end maturation (*Maurer et al., 2014*), (ii) acceleration of tubulin dissociation from the microtubule end (compare *Figure 5A* and *Figure 2D*), possibly due to effects of EBs on the microtubule end structure ('taper' or 'sheet') during growth (*Vitre et al., 2008*), and (iii) direct structural destabilisation of the cap (i.e. increase of the end density threshold) (*Figure 6C*), which agrees with recent structural studies indicating that EBs induce strain in the microtubule lattice (*Zhang et al., 2015*). These findings further support the view that end binding proteins of the EB1 family may have evolved to modulate the stability of the functionally essential protective end structure of the microtubule and may then have gained the additional function to recruit a variety of other unrelated plus end binding proteins (+TIPs), whereas in parallel more effective, for example ATP-dependent modulators of cap stability such as depolymerases of the kinesin-13 family evolved (*Akhmanova and Steinmetz, 2015*; *Brouhard and Rice, 2014*; *Duellberg et al., 2013*; *Gardner et al., 2011b*; *Howard and Hyman, 2007*; *Maurer et al., 2012*).

## Outlook

While the debate about the size of the stabilising cap was ongoing, microtubule end binding proteins were discovered (*Beinhauer et al., 1997*; *Su et al., 1995*) and later used to facilitate microtubule end tracking in living cells (*Busch and Brunner, 2004*; *Komarova et al., 2009*; *Mimori-Kiyosue et al., 2000*; *Seetapun et al., 2012*). Our quantitative demonstration that the EB binding region is the protective microtubule cap means that EBs can be used to monitor the momentary stability of individual microtubules. This will be useful for the further investigation of the momentary microtubule stability in reconstituted systems, for example for the study of its dependence on the microtubule growth time, but also for studies of how microtubule stability is modulated in living cells in real time. For the future, such experiments combined with quantitative analysis promise to yield a deeper mechanistic understanding of how microtubule ageing, and the presence of regulatory proteins and drugs control microtubule dynamics.

# Materials and methods

## Proteins

Mal3-GFP was expressed and purified as described (*Maurer et al., 2011*). Porcine tubulin was purified (*Castoldi and Popov, 2003*) and labelled with biotin (Pierce) or Alexa568 (Life technologies) using NHS esters following standard protocols (*Hyman et al., 1991*).

## Microfluidic device fabrication

Microfluidic channels with a functionalised glass surface were fabricated combining glass surface chemistry (*Bieling et al., 2010*) and soft lithography techniques (*Whitesides et al., 2001*). The overall design was inspired by a recent study on actin filaments (*Jegou et al., 2011*). 24 mm x 60 mm glass coverslips (Menzel) were covalently passivated (against non-specific protein adsorption) with polyethylene glycol (PEG) and functionalised with biotin as described (*Bieling et al., 2010*). Silicon moulds with negative channel patterns were produced by deep reactive ion etching (*Holmes et al., 2014*). Channel dimensions are detailed in *Figure 1—figure supplement 1A*.

A mixture of polydimethylsiloxane (PDMS) and curing agent (both from Dow Corning; 10:1 ratio w/w) was poured over a mould, degassed for 2 hr at 4°C (Modulyo 4K, Edwards) and polymerised for ~12 hr at 68°C. The structured side of the peeled-off PDMS elastomer and of the functionalised surface of the glass coverslip were treated with air plasma (Diener, Femto) for 42 s. A small area of 4 mm x 8 mm, where imaging was to occur, was protected from plasma radiation by a small PDMS block to ensure the integrity of the surface functionalization. Immediately after plasma treatment the exposed sides were bonded to form the channels, and holes for the inlet and outlet channels were created using biopsy punchers (Harris Uni-Core I.D. 0.75 mm). Tubing (Tygon; diameter: 0.5 mm) was connected to the PDMS inlets/outlet via short metal extensions (21 gauge hollow cylinders, cut from 'microlances' Becton Dickinson) and Hamilton gas tight syringes (500 or 1000 μl total volume). The microfluidic devices were used for TIRF microscopy experiments immediately after assembly.

## TIRF microscopy-based tubulin washout assay with microfluidic control

All experiments were performed using a previously described TIRF microscopy set-up at a constant temperature at 30°C (*Duellberg et al., 2014*; *Maurer et al., 2014*). Fast solution exchange in the micro-channel was achieved by switching the flow from three different inlets that were controlled by independent syringe pumps (Aladdin, World Precision Instruments) and manual valves (Cole-Parmer) between pumps and inlets. The microfluidic set-up and all solutions were pre-warmed to 30°C just before the experiment. To assemble a sample, short GMPCPP-stabilised, Alexa568 and biotin-labelled microtubule 'seeds' were introduced through one inlet and allowed to attach to the functionalised glass surface via neutravidin (*Bieling et al., 2010*). In brief, channels were purged with 100 μl assay buffer, 30 μl assay buffer supplemented with neutravidin (Life Technologies, 0.05 mg/ml), 100 μl assay buffer, 30 μl GMPCPP seeds in assay buffer and again 100 μl assay buffer using always the same inlet.

To initiate microtubule growth, 10–35 μM Alexa568-labelled tubulin (labelling ratio was always 12.5%) in final imaging buffer with or without 200 nM Mal3-GFP was introduced through the second inlet at a constantly maintained flow rate of 15 μl/min. Unless stated otherwise, the final imaging buffer was 80 mM K-PIPES (pH 6.85), 1 mM EGTA, 1 mM dithiothreitol, 90 mM KCl, 0.5 mM $MgCl_2$, 5 mM 2-mercaptoethanol, 10 mM ascorbic acid, 0.1% (w/v) methylcellulose, 2 mM GTP, 50 μg/ml β-casein, 20 mM glucose, 0.25 mg/ml catalase, 0.5 mg/ml glucose oxidase. For data presented in *Figure 6A*. the $MgCl_2$ concentration was increased to 4 mM.

Microtubules were allowed to grow for ~100 s until tubulin was quickly washed out by switching the flow to the third inlet leaving all other buffer constituents unchanged. Alexa568-microtubules and Mal3-GFP images were recorded simultaneously in separate channels at a frame rate of 4 Hz with 100 ms exposure time per image, unless stated otherwise.

For some controls, the tubulin concentration was changed from 12 to 25 μM and vice versa (*Figure 1—figure supplement 1D*) or the tubulin concentration was kept constant at 15 μM while 75 nM Mal3-GFP was washed in and out to demonstrate that solution exchange does not interfere with microtubule growth or cause catastrophes by itself (*Figure 1—figure supplement 1C*)

## Generation of microtubule end position tracks, Mal3-GFP end intensity time series and tubulin background time series

Growing microtubule ends were automatically tracked using a previously described script (*Maurer et al., 2014*; *Ruhnow et al., 2011*). Briefly, microtubules were coarsely identified by the user at the start of each movie, including microtubule seed position and polarity. A two-dimensional model was then automatically fit to the intensity data for the microtubule plus-end in all subsequent frames. For each tracked microtubule, this gave the position of the microtubule end at every time point. From simulations, the tracking precision is estimated to be 20–30 nm (*Bohner et al., 2015*) at the typical signal-to-noise levels of >3 determined for these movies. When present, the simultaneously imaged Mal3-GFP intensity was quantified at the microtubule end position, and the lattice intensity was quantified 1.5 μm from the end along the microtubule. To generate soluble tubulin background time series, the spatially averaged background intensity of soluble Alexa568-tubulin was determined during the fitting procedure of the microtubule end, at every time point.

## Microtubule track and Mal3-GFP intensity analysis after tubulin washout

The microfluidic channel was aligned with the camera axis before imaging. Only plus ends of microtubules that were aligned with the flow direction along the x-axis and did not show appreciable bending were considered for analysis to avoid potential mechanical effects on microtubule dynamic behaviour due to bending (*Schaedel et al., 2015*). For each microtubule, the delay time between tubulin washout and catastrophe, the corresponding depolymerisation length and the kinetic parameters characteristic of its dynamics were quantified using a fully automated custom script (Matlab, Mathworks), as follows.

Firstly, the washout time point was identified from the inflection point of an error-function fit to the background intensity of the tubulin fluorescence channel (top panel in *Figure 1D* and *Figure 1—figure supplement 1B*). The average time for 90% buffer exchange (5% - 95%) was 199 (± 32 s.e.m) ms, as determined from 9 washout profiles measured at 7.7 Hz (*Figure 1—figure supplement 1B*).

The pre-washout instantaneous microtubule growth speed was found from a linear fit to the end position data over a 10 s time window exactly before washout. After washout, the catastrophe time was found by fitting the position data with the integral of the following function for the speed of the shrinking microtubule end (*Figure 1C*, middle panel)

$$0.5(v_1 + v_2) + 0.5(v_1 - v_2)\mathrm{erf}[(t_0 - t)/\sqrt{2}\sigma]$$

This describes a transition between a period of slow linear shrinkage $v_1$ to a period of fast depolymerisation $v_2$, through an error function centred at $t_0$, with transition width $\sigma$. The catastrophe point was defined as the time at which there was a 25% change between the slow shrinkage and the fast depolymerisation phase (*Figure 1D*, bottom panel, *Figure 2—figure supplement 1A*). This point was found to be a sensible empirical measure for the fitted microtubule end trajectory visibly departing from the noise of the linear slow shrinkage phase between tubulin washout and catastrophe (*Figure 2—figure supplement 1B*). Thence the measured delay time was defined as the difference between the washout and catastrophe times, and the measured shrinkage length defined as the corresponding difference in fitted microtubule end positions (*Figure 2E*, *Figure 2—figure supplement 1B*). A linear approximation for the slow shrinkage speed between washout and catastrophe $v_s$ was obtained from a linear fit to this part of the track. $v_2$ was directly taken as the fast depolymerisation speed after catastrophe $v_f$.

Spearman's rank coefficient was used to examine correlations between the above parameters as well as the Mal3-GFP end and lattice intensities at tubulin washout and catastrophe. To investigate the importance of measuring growth speeds and Mal3-GFP end intensities 'instantaneously' at washout for the magnitude of their correlation with the delay times, we also determined growth velocities and Mal3-GFP intensities over 10 s and 1 s time windows, respectively, at various times before washout. All reported correlation coefficients are summarised in the *Supplementary file 1C*.

For each assay condition, plots showing the average microtubule end position around catastrophe were made by aligning all individual tracks at their determined catastrophe time and catastrophe end position, followed by resampling and averaging the data. Corresponding average Mal3-GFP intensity plots were made by aligning individual Mal3-GFP intensity profiles at the catastrophe time, followed by resampling and averaging the data.

## Analysis of microtubule growth orientation

For each tracked microtubule, the 2D fit in the tracking procedure gave the orientation of the microtubule end region at every time point, with the positive x-axis of the image defined as zero degrees. The pre-washout average orientation and corresponding standard deviation were calculated over a 10 s time window just before washout, along with the subsequent delay time until catastrophe. For each assay condition, scatter plots of delay times versus the orientations of the microtubule end regions were produced. For analysis of statistical correlations, the magnitude of the orientation relative to the mean was used.

## Standard dynamic microtubule TIRF microscopy assay

To compare the Mal3-GFP intensity variability at growing microtubule ends (*Figure 4—figure supplement 1*) in the presence of GTP and on GTPγS microtubules, microtubules growing in the

presence of 22 µM Alexa568-labelled tubulin and 75 nM Mal3-GFP were imaged in standard flow chambers consisting of one functionalised and one passivated glass separated by double-sided sticky tape, as described previously (*Bieling et al., 2010*). The chambers were filled manually; other conditions were as above, except that the final assay buffer was supplemented with 3.5 mM MgCl₂, did not contain ascorbic acid and contained either 1 mM GTPγS (Roche) or 1 mM GTP (Fermentas). Microtubule end positions and Mal3-GFP intensities were recorded and analysed as detailed in the previous section.

To compare these data sets, we determined the relative variance, *D*, of GFP intensities, where *D* is the variance divided by the mean. To confirm that the broad distribution of Mal3-GFP signals at microtubule ends reflects differences in the number of high affinity binding sites, we compared microtubules grown in the presence of GTP and GTPγS, the latter being a GTP or GDP/Pi analogue with very slow hydrolysis kinetics which mimics a microtubule end with a roughly constant number of EB binding sites (*Maurer et al., 2011*). For each microtubule ($n = 9$, from 3 independent experiments, at least 1500 frames per microtubule), GFP intensities were measured for each frame and the mean and variance were determined.

## Comet analysis

To determine the conformational maturation rates at microtubule ends, Mal3-GFP intensity profiles ('comets') were analysed (*Maurer et al., 2014*). For tubulin washout experiments in the presence of 200 nM Mal3-GFP, the growth episodes before washout were directly used for analysis. To obtain estimates for maturation rates in washout experiments without Mal3-GFP, we performed independent experiments under essentially identical conditions as in tubulin washout experiments without Mal3-GFP, however with added low 'spike' concentrations of 0.75 nM or 1 nM Mal3-GFP, as indicated in the *Supplementary file 1A,* i.e. concentrations well below the $K_d$ of Mal3-GFP binding to microtubule ends (*Maurer et al., 2011*).

To generate averaged Mal3-GFP comets, speed-sorted subsets of microtubule tracks were generated. Images of the microtubule end and corresponding Mal3-GFP signal were cropped from each movie frame and aligned at the detected end position: average images were produced from a total of ~5500 individual image frames, from 38 separate microtubules (200 nM Mal3-GFP condition). Average one-dimensional Mal3-GFP comet profiles were generated and analysed as described (*Maurer et al., 2014*). For the growth speed range of the microtubules used for comet analysis here, a simple kinetic 'one-step' model of microtubule maturation (*Maurer et al., 2014*) describes the distribution of EB binding sites sufficiently well: a microtubule growing at speed $v_g$ is assumed to form EB binding sites approximately instantly, which subsequently mature into weak (lattice) binding sites at a rate $k_m$. Hence the microtubule will have an approximately mono-exponential decay of binding sites in space $x$, giving a spatial profile of the linear density of all types of binding sites

$$n(x) = n_{x0}e^{-xk_m/v_g} + n_{lat} \tag{1}$$

where $n_{x0}$ is the maximal binding site density, $v_g/k_m = L_{comet}$ is the decay length of the binding region or the 'comet length', and $n_{lat}$ is a step function representing a constant background signal due to lattice binding (the affinity of lattice binding sites is roughly ten-fold reduced compared to end binding sites [*Maurer et al., 2011*]).

The measured average fluorescent intensity profile, $I(x)$, is a convolution of the binding site distribution, $n(x)$, with a Gaussian function, $g(x)$, that accounts for the optical PSF of the microscope and effects of averaging

$$g(x) = e^{-\frac{1}{2}\left(\frac{x-x_c}{\sigma}\right)^2}$$

$$I(x) = g(x) * n(x) \tag{1a}$$

where $\sigma$ is the width of the Gaussian, and $x_c$ accounts for any spatial offset of the start of the EB binding sites from the fitted microtubule end position.

*Supplementary file 1A* summarises the kinetic rates found from the average comets for each examined condition.

## EB cap decay kinetics after washout and threshold models

We consider here the evolution of the number of the high affinity EB binding sites or their density at the end of the microtubule. We assume here that a microtubule with $n_{pf}$ protofilaments, each composed of tubulin dimers of length $l_{dim}$ behaves as a single one-dimensional filament with an effective linear density of potential binding sites of $n_{x0} = n_{pf}/l_{dim}$. We further assume that at the moment of tubulin washout (t = 0), a microtubule with growth velocity $v_g$ and maturation rate has an approximately mono-exponential profile of EB binding sites in space with

$$n(x) = n_{x0}e^{-xk_m/v_g} \tag{1b}$$

Hence, integrating [1b] along the whole microtubule, the total number of EB binding sites at washout is

$$N(t=0) = \int_0^\infty n_{x0}e^{-xk_m/v_g}dx = n_{x0}v_g/k_m = n_{x0}L_{comet} \tag{2}$$

As **Figure 5B** illustrates, after removal of free tubulin, all binding sites decay in time independently at a rate $k_m$ as a consequence of continued maturation giving (in the absence of shrinkage) a spatio-temporal profile

$$n(x,t) = n_{x0}e^{-xk_m/v_g}e^{-k_mt} \tag{3}$$

Additionally binding sites (and non-binding site tubulins) are lost from the microtubule end due to shrinking at a speed $v_s$ (see **Figure 5B**). Defining the shrinking end position as $x'_0 = v_st$, the total number of sites at a given time after washout is

$$N(t) = \int_{v_st}^\infty n_{x0}e^{-xk_m/v_g}e^{-k_mt}dx = N(0)e^{-k_m(v_s/v_g+1)t} \tag{4}$$

The measured Mal3-GFP intensity (corrected for background intensity) is proportional to this total number of binding sites. Thus, from [4] the predicted temporal decay of the Mal3-GFP intensity is

$$I_{Mal3}(t) = I_{Mal3}(0)e^{-k_{Mal3}t} \tag{5}$$

with

$$k_{Mal3} = k_m(v_s/v_g + 1) \tag{6}$$

as used for the analysis in **Figure 5A**.

Hence, the average time to catastrophe can be estimated (**Figure 5D** left) using

$$\langle T_{Mal3}\rangle = \frac{-\ln[\langle I_{Mal3}(cat)\rangle / \langle I_{Mal3}(0)\rangle]}{k_{Mal3}} \tag{7}$$

Assuming that the dispersion of delay times depends mostly on the variation of the size of EB binding site caps at washout and catastrophe, a rough estimate for the dispersion of delay times (**Figure 5D** left) can be produced in two steps:

(i) The measured dispersion of the Mal3-GFP end signals is a result of measurement noise and the true variability of cap sizes. To extract an estimate for the intrinsic standard deviation of the number of EB binding sites $\sigma_{EB}$, we assume that the standard deviation of the Mal3-GFP signal on the microtubule lattice $\sigma_{lattice}$ can be considered a rough estimate of the measurement noise. Hence the intrinsic standard deviation of the EB sites at washout and at catastrophe is approximately

$$\sigma_{EB} = \sqrt{\sigma_{Mal3}^2 - \sigma_{lattice}^2}$$

(ii) $k_{Mal3}$ is calculated from $k_m$, $v_s$, $v_g$ assuming that their errors are small compared to the dispersion of cap sizes. The standard deviations of the Mal3-GFP intensities at washout and catastrophe are then propagated according to standard rules of error propagation.

The expected mean shrinkage length is

$$\langle L_{shrink}\rangle = v_s\langle T_{Mal3}\rangle \tag{8}$$

and a rough estimate for its standard deviation is obtained by error propagation of the standard deviation of $T_{Mal3}$ (**Figure 5D** right).

## Kinetic threshold models based on terminal density or entire cap size

Assuming 13 protofilaments and 8 nm tubulin dimers, i.e. $n_{x0} = 13/8\text{nm}$, the temporal decay of the linear density of EB binding sites at the end of a shrinking microtubule, $x_0'$ (called here the end density) can be found by setting $x = x_0' = v_s t$ in [3]:

$$n(x_0', t) = (13/8\text{nm})\, e^{-k_m(v_s/v_g+1)t} \tag{9}$$

Thus the decay rate of the linear density of EB sites at the microtubule end is also $k_{Mal3}$.

For an <u>end density threshold</u> model (**Figure 6B,C**), we assume that catastrophe occurs when $n(x_0', t)$ reaches a critical (always constant) end density $n_{\text{crit}}$ after a delay time $T_{end}$:

$$n_{crit} = (13/8\text{nm})e^{-k_m(v_s/v_g+1)T_{end}} \tag{10}$$

which gives a predicted delay time

$$T_{end} = -\frac{\ln(n_{crit}/(13/8\text{nm}))}{k_m(v_s/v_g + 1)} = -\frac{\ln(f_{crit})}{k_{Mal3}} \tag{11}$$

with a fractional end threshold $f_{crit}$.

For a <u>total number threshold</u> model (**Figure 6A**, dashed line), we assume that a catastrophe occurs when the total number of sites along the whole microtubule reaches a critical (always constant) number $N_{\text{crit}}$ after a delay time $T_{tot}$. From [4],

$$N_{crit} = \text{N}(0)e^{-k_m(v_s/v_g+1)T_{tot}} = (13/8\text{nm})(v_g/k_m)e^{-k_m(v_s/v_g+1)T_{tot}} \tag{12}$$

this gives a delay time

$$T_{tot} = -\frac{\ln[N_{crit}/N(0)]}{k_m(v_s/v_g + 1)} = -\frac{\ln[N_{crit}(k_m/v_g)/(13/8\text{nm})]}{k_m(v_s/v_g + 1)} \tag{13}$$

For both models, an estimate for the shrinkage length between washout and catastrophe is then obtained as

$$L_{shrink} = v_s T_{end/tot} \tag{14}$$

Comparing [11] and [13], the two models predict different dependencies on the growth speed $v_g$; hence, fitting the measured delay times over a large range of growth speeds can be used to distinguish between both models (**Figure 6A**).

To fit the data in **Figure 6A** using **Equation 11,13**, the empirical relationship between $v_s$ and $v_g$ was determined from a linear fit over the large speed range explored, giving $v_s = 0.28v_g + 22$ (**Figure 6—figure supplement 1C**). To fit the data in **Figure 6C**, the mean values of $v_s = 38$ nm/s and 28 nm/s were used for the data set with 200 nM Mal3-GFP and without Mal3-GFP, respectively.

## Threshold model considering the number of binding sites within a specific length (L)

This 'L-model' (**Figure 6D**) considers the total number of EB binding sites within a constant length $L$ from the end of the microtubule, which is shrinking at speed $v_s$.

$$N_L(t) = N_L(0)e^{-k_m(v_s/v_g+1)t} \tag{15}$$

where the number of sites within $L$ at $t = 0$ is given by

$$N_L(0) = \left(\frac{13}{8\text{nm}}\right)(v_g/k_m)[1 - e^{-k_m L/v_g}] \tag{16}$$

Here we assume a catastrophe occurs when the number of sites within a length $L$ from the end reaches a critical number $N_{Lcrit}$ after a delay time $T_L$, which gives a delay time

$$T_L = -\frac{\ln[N_{Lcrit}/N_L(0)]}{k_m(v_s/v_g + 1)} = -\frac{ln[N_{Lcrit}(8\text{nm}/13)(k_m/v_g)(1 - e^{-k_m L/v_g})^{-1}]}{k_m(v_s/v_g + 1)} \tag{17}$$

The growth speed dependence of the Mal3-GFP intensities at washout and catastrophe can be used to constrain a global fit of the speed-dependent delay times (*Figure 6D*, *Figure 6—figure supplement 2*). Expressions for the Mal3-GFP intensity at washout (t = 0) and for the catastrophe time (t = $T_L$, here using the *L*-model) can be derived from [4]:

$$\text{At washout}: \ I_{Mal3}(0) = F v_g \tag{18}$$

$$\text{At catastrophe}: \ I_{Mal3}(T_L) = F N_{Lcrit}(8\text{nm}/13) k_m (1 - e^{-k_m L/v_g})^{-1} \tag{19}$$

where $F$ is a constant proportionality factor dependent on camera parameters etc.

## Threshold models for 2-step microtubule end maturation

For simplicity, we have so far approximated the density distribution of the EB binding sites by a mono-exponential decay, as described above. We previously reported evidence for a more complex distribution suggesting a delay between tubulin addition and EB binding site generation (*Maurer et al., 2014*). This delay could be described by an additional kinetic step with a rate $k_1$ that is fast compared to the decay rate, $k_m$, of the EB binding sites. This 2-step kinetic model leads to modified expressions for the threshold models:

Assuming a kinetic process $A \xrightarrow{k_1} B \xrightarrow{k_m} C$, where *A* is a non-binding but stabilizing end state, *B* is the EB binding state, and *C* is the mature lattice state, one finds the time course after washout of the <u>end density of the sum of A and B sites:</u>

$$n_{A+B}(x_0', t) = (13/8\text{nm})/(k_1 - k_m)[k_1 e^{-k_m(v_s/v_g + 1)t} - k_m e^{-k_1(v_s/v_g + 1)t}] \tag{20}$$

Setting $t = T_{end2}$ (the delay time according to the 2-step end density threshold model) and $n_{A+B}(x_0', T_{end2}) = n_{crit2}$ (the critical threshold density required for microtubule stability according to the 2-step threshold model), we performed a fit to the data solving $T_{end2} = f(k_1, k_m, v_s, v_g, n_{crit2})$ numerically using Matlab.

For the time course of the <u>total number of all A and B sites</u> one finds

$$N_{A+B}(x_0', t) = (13/8\text{nm}) v_g/(k_1 - k_m)[(k_1/k_m)e^{-k_m(v_s/v_g + 1)t} - (k_m/k_1)e^{-k_1(v_s/v_g + 1)t}] \tag{21}$$

As above, setting $t = T_{tot2}$ and $N_{A+B}(x_0', T_{tot2}) = N_{crit2}$, we performed a fit to the data solving $T_{tot2} = f(k_1, k_m, v_s, v_g, N_{crit2})$ numerically using Matlab.

Comparing *Equation 20 and 9*, and *Equation 21 and 4*: for $k_1 \gg k_m$ the expressions for the end density and total number of A and B sites for a 2-step maturation process transform into the expressions of the simpler models used throughout this study that assume only one maturation step with rate constant $k_m$.

### Numerical data

Numerical values for the data presented in *Figures 4*, *6A and C* are listed on corresponding sheets in *Supplementary file 2*.

### Software

All custom software and scripts mentioned in the text are available from the authors on request.

## Acknowledgements

We thank T Fallesen for help with improving the mechanical stability of the experimental set up. We thank G Bohner, EL Grishchuk and J Rickman for helpful discussions. We thank N Goehring, J Rickman, J Roostalu, G Salbreux and E Schnur for critically reading the manuscript. This work was supported by the Francis Crick Institute which receives its core funding from Cancer Research UK, the UK Medical Research Council, and the Wellcome Trust. In addition, this research has received

funding from the European Union's Seventh Framework Programme under ERC grant agreement No 323042.

## Additional information

### Funding

| Funder | Author |
| --- | --- |
| European Research Council | Thomas Surrey |
| Cancer Research UK | Thomas Surrey |

The funders had no role in study design, data collection and interpretation, or the decision to submit the work for publication.

### Author contributions

CD, Conception and design, Acquisition of data, Analysis and interpretation of data, Drafting or revising the article, Contributed unpublished essential data or reagents; NIC, Derived equations for threshold models, analyzed data and wrote the manuscript, Analysis and interpretation of data, Drafting or revising the article; DH, Contributed unpublished essential data or reagents; TS, Derived equations for threshold models, Conception and design, Analysis and interpretation of data, Drafting or revising the article

### Author ORCIDs

Thomas Surrey, http://orcid.org/0000-0001-9082-1870

## Additional files

**Supplementary files**

• Supplementary file 1. (A) Fit parameters from analysis of comet profiles. (B) Fit parameters for the various different threshold models. (C) Correlations between measured parameters.

• Supplementary file 2. Numerical values for the data presented in *Figures 4*, *6A and C*.

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
