## [Decision Letter]

Thank you for submitting your work entitled "The size of the EB cap determines instantaneous microtubule stability" for consideration by *eLife*. Your article has been reviewed by three peer reviewers, and the evaluation has been overseen by a Reviewing Editor and John Kuriyan as the Senior Editor.

The reviewers have discussed the reviews with one another and the Reviewing Editor has drafted this decision to help you prepare a revised submission.

Summary:

The authors employed microfluidics to monitor microtubule catastrophe induced by a sudden removal of free tubulin. A unique advantage of this approach is that all major parameters of microtubule dynamics (growth rate, duration of the pause between the removal of tubulin and the catastrophe, depolymerization rates) are obtained for specific microtubules that are continuously monitored throughout the experiment. The quality of the data is excellent. The main conclusion is that microtubules are protected by a long cap delineated by the binding of EB. The length of the cap positively correlates with the growth rate and the ability of microtubule to remain stable in the absence of free tubulin. Importantly, the data presented in the manuscript suggest that these correlations have not been observed in previous studies due to rapid fluctuations, the size of the cap needs to be measured right before the catastrophe. This is an excellent paper whose publication in *eLife* is strongly recommended.

Essential revisions:

1) All three reviewers agree that the issue of mechanical stress on microtubule stability, brought to general attention by the work of Manuel Thery and co-workers (see Schaedel et al., Nature Materials 2015) deserves more attention. One reviewer notes that the authors cite the Thiery paper in the Methods and states that the authors only analyze microtubules aligned with the direction of the flow to avoid analyzing damaged microtubules, but also notes that it is unclear what is the tolerance for "aligned with the flow", and wonders if the authors can completely exclude mechanical stress, given that they need very rapid fluid exchange. All reviewers agree that the issue of mechanical stress should be addressed directly in the Results, not only the Methods. Furthermore, the reviewers agree that if there was an appreciable variability in the angular orientation of the microtubules, one way to address this concern would be to present data that is binned according to the angle relative to the direction of flow.

2) The effects of microtubule ageing on catastrophe frequency has been described (Gardner et al., Cell 2011), and competing models to explain this phenomenon have been proposed (Coombes et al., Curr Biol 2013, Zakharov et al., Biophys J 2015, and Bowne-Anderson, Bioessays 2013). The authors do not address this point in their manuscript, but the reviewers agree that they should do it, at least in writing, and ideally could also report whether their data can provide additional evidence to evaluate this model, and in particular with reference to the question whether the delay times correlate with the age of the microtubule. No additional experiments are required, but rather an analysis of the existing data to address this point and a discussion of the previous literature.

3) The reviewers strongly recommend that the authors tabulate all of their quantitative measurements in an appendix. There have been a number of attempts to develop computational/kinetic models for catastrophe, and different groups often refine their model against different experimental datasets. This causes a lot of unnecessary confusion/conflict. Recapitulating the comprehensive measurements reported here will provide a stringent new challenge for models.

4) Finally, one of the reviewers noted that the models discussed in the manuscript treat the microtubule end in a simplified manner. The reviewer writes: "If I understand the model correctly, the EB binding region extends to the very end of the microtubule. This is contrary to previous work from the Surrey group which showed that EB's do not bind to the very end of the microtubule (Maurer et al., 2014). In the Maurer paper, the tip was modeled as moving from state A (non-EB binding) state B (EB binding)  state C (post-hydrolysis, non-EB binding). The reason this matters in this paper is because of the shrinkage of the tip after the tubulin is washed out. Is this shrinkage reducing the number of EB binding sites? Well, maybe not, because there is no EB in the region that is shrinking (Figure 5, mean shrinkage length of 165 nm, compared to the size of the A region in Maurer 2014, approximately 100 nm). Does the use of a two-step maturation model weaken or strengthen the conclusions presented here?". A clarification of this point in your rebuttal, and any useful editing that might help clarifying this point, would be useful.

---

## [Author Response]

*Essential revisions: 1) All three reviewers agree that the issue of mechanical stress on microtubule stability, brought to general attention by the work of Manuel Thery and co-workers (see Schaedel et al., Nature Materials 2015) deserves more attention. One reviewer notes that the authors cite the Thiery paper in the Methods and states that the authors only analyze microtubules aligned with the direction of the flow to avoid analyzing damaged microtubules, but also notes that it is unclear what is the tolerance for "aligned with the flow", and wonders if the authors can completely exclude mechanical stress, given that they need very rapid fluid exchange. All reviewers agree that the issue of mechanical stress should be addressed directly in the Results, not only the Methods. Furthermore, the reviewers agree that if there was an appreciable variability in the angular orientation of the microtubules, one way to address this concern would be to present data that is binned according to the angle relative to the direction of flow.*

As suggested, we have re-examined our existing data. Our fitting software allows us to extract the orientation of the analysed microtubules. Although our experiment was designed in a way that microtubules are aligned with the flow during growth to minimise mechanical stress of the microtubules, we nevertheless quantified the variability in the orientation of the microtubule end regions with respect to the flow direction (the flow cells were manually aligned so that their long axis was parallel to the x-axis of the camera chip). For all three data sets presented in the manuscript we determined the angles of microtubule orientation and found that they were within +/-10 degrees of the flow direction. Plotting the delay times as a function of orientation for all data sets demonstrated convincingly that there was no correlation between angle and delay time. Any potential mechanical strain introduced during fluid exchange is expected to be larger for larger angles; hence, we can conclude that within the range of microtubule orientations of our experiment, there is no evidence of mechanical strain affecting the observed momentary microtubule stabilities (delay times between tubulin washout and catastrophe). We present scatter plots of the delay times vs. the 'raw' orientations for all datasets in Figure 1—figure supplement 2 and added a corresponding description of this new analysis to the Results, to the Methods and to the legends. For the benefit of the reviewers, we also add scatter plots for the tubulin concentration dataset (Figure 6) showing the delay times vs. the magnitude of the average orientations (Figure 7), and of the delay times vs. the standard deviations of the orientation (Figure 7) (each orientation is an average over 10 s before tubulin washout). All scatter plots show that there are no correlations, demonstrating that the observed differences in delay times are not caused by mechanical stress. We thank the reviewers for suggesting this analysis and consider this an interesting new result that has strengthened our conclusions.

Author response image 1.(**A**) Delay time vs. the magnitude of the orientation of the data in Figure 1—figure supplement 2 relative to the mean: for each data point of a specific tubulin concentration, the mean of the distribution has been subtracted and the magnitude of this relative orientation is plotted.Statistical analysis shows no significant correlation (Pearson’s *r* =0.01). (**B**) Delay time vs. the standard deviation of the growth orientation measured over 10s before washout. Statistical analysis shows no significant correlation (Pearson’s *r* =0.1, p = 0.17).**DOI:**
http://dx.doi.org/10.7554/eLife.13470.021

*2) The effects of microtubule ageing on catastrophe frequency has been described (Gardner et al., Cell 2011), and competing models to explain this phenomenon have been proposed (Coombes et al., Curr Biol 2013, Zakharov et al., Biophys J 2015, and Bowne-Anderson, Bioessays 2013). The authors do not address this point in their manuscript, but the reviewers agree that they should do it, at least in writing, and ideally could also report whether their data can provide additional evidence to evaluate this model, and in particular with reference to the question whether the delay times correlate with the age of the microtubule. No additional experiments are required, but rather an analysis of the existing data to address this point and a discussion of the previous literature.*

Our experiments were designed to keep ageing effects at a minimum. Tubulin washout was always performed at a roughly constant time of ~100s after flowing in the tubulin. We considered this restriction necessary given the already complex and rich set of data. Therefore, our model does not describe ageing effects, since all microtubules have roughly the same age. We state this more clearly now in the Discussion. We state that the fast tubulin washout experiments as performed here will be an interesting and powerful method to test different ageing models. We have begun with such experiments and analysis, and given the amount of additional data needed, we conclude that this will be a study of its own in the future, going well beyond the scope of this study here.

*3) The reviewers strongly recommend that the authors tabulate all of their quantitative measurements in an appendix. There have been a number of attempts to develop computational/kinetic models for catastrophe, and different groups often refine their model against different experimental datasets. This causes a lot of unnecessary confusion/conflict. Recapitulating the comprehensive measurements reported here will provide a stringent new challenge for models.*

As requested, we have tabulated our data in the appendix.

4) Finally, one of the reviewers noted that the models discussed in the manuscript treat the microtubule end in a simplified manner. The reviewer writes: "If I understand the model correctly, the EB binding region extends to the very end of the microtubule. This is contrary to previous work from the Surrey group which showed that EB's do not bind to the very end of the microtubule (Maurer et al., 2014). In the Maurer paper, the tip was modeled as moving from state A (non-EB binding) state B (EB binding)  state C (post-hydrolysis, non-EB binding). The reason this matters in this paper is because of the shrinkage of the tip after the tubulin is washed out. Is this shrinkage reducing the number of EB binding sites? Well, maybe not, because there is no EB in the region that is shrinking (Figure 5, mean shrinkage length of 165 nm, compared to the size of the A region in Maurer 2014, approximately 100 nm). Does the use of a two-step maturation model weaken or strengthen the conclusions presented here?". A clarification of this point in your rebuttal, and any useful editing that might help clarifying this point, would be useful.

In our previous version of the manuscript we mentioned in the Results section that applying the 2-step model did not change the quality of the fits to our data or any of our main conclusions, but can change the predicted threshold values. In Figure 6—figure supplement 3, we now show these fits applying the 2-step model to all three datasets in comparison to the fits using the 1-step model. Importantly, using the 2-step model confirms the conclusion that the *end density* and not the *total number* of binding sites defines the threshold of stability. Two models make similar predictions, essentially because the first (A) region is typically short, see Maurer et al., Curr Biol. 2014. We extended the description of the 2-step model fits in the Results section and added a section to the Methods explaining how the analysis was performed. We feel that the manuscript is more accessible if we continue to focus on the 1-step model throughout and explain under which conditions the predicted thresholds need to be adjusted when the 2-step model is applied. These adjustments are rather minor, essentially because in the 2-step maturation model the first step is always faster than the second step, especially in the presence of EB proteins, as we found previously (Maurer et al., Curr Biol 2014). This means that under more physiological conditions (presence of EBs), the first step of the 2-step model can be neglected, leaving us with a 1-step model. In the absence of EBs (only relevant for some in vitro experiments), we found previously for very fast growing microtubules that the first step is predicted to be ~5x faster than the second step. In this case (absence of EBs), the 2-step model predicts ~25% higher threshold values of stability in the tubulin washout experiment than the 1-step model does. Conceptually, we do not consider this change as very important, especially given that this range is close to the experimental errors and/or observed variability in the data. Nevertheless, we agree that it is important to clarify these points.